# Vaginal hormone-free moisturising cream is not inferior to an estriol cream for treating symptoms of vulvovaginal atrophy: Prospective, randomised study

**Susana Garcia de Arriba**[1]*, **Lisa Grüntkemeier**[1], **Manuel Häuser**[1], **Theodor W. May**[2], **Clarissa Masur**[1], **Petra Stute**[3]

1 Dr. August Wolff GmbH & Co. KG Arzneimittel, Bielefeld, Germany, 2 Society for Biometrics and Psychometrics GbR, Bielefeld, Germany, 3 Department of Obstetrics and Gynaecology, Inselspital University Clinic of Bern, Bern, Switzerland

* susana.garciadearriba@drwolffgroup.com

## Abstract

This prospective, open-label, multicentre, multinational, randomised trial investigated the non-inferiority of treatment with a vaginal hormone-free moisturising cream compared to a vaginal estriol (0.1%) cream in a panel of post-menopausal women suffering from symptoms of vulvovaginal dryness in a parallel group design. In total, 172 post-menopausal women were randomly allocated to either one of the two treatments, each administered for 43 days. The primary endpoint was the total severity score of subjective symptoms (dryness, itching, burning and pain unrelated to sexual intercourse) of the respective treatment period. Secondary endpoints were severity of single subjective symptoms (including dyspareunia if sexually active), impairment of daily life, Vaginal Health Index, as well as assessment of safety. In both groups, women treated with hormone-free moisturising cream and those treated with estriol cream, total severity score improved significantly compared to baseline by 5.0 (from 6.1 to 1.1) and by 5.4 (from 6.0 to 0.6), respectively, after 43 days of treatment (p < 0.0001). One-sided test of baseline differences (for a clinically relevant difference Δ = 1.5) confirmed the hormone-free moisturising cream to be non-inferior to the estriol cream. Severity of dyspareunia as well as impairment of daily life due to subjective symptoms, significantly improved for both treatment groups (p<0.0001). Subgroup analysis of women with mild or moderate impairment of daily life at baseline caused by "vaginal dryness" symptoms benefited from both creams, while women with severe impairment showed a significantly greater benefit from the estriol cream (p = 0.0032). Both treatments were well tolerated with no serious adverse events occurring. This study provides clinical evidence that a hormone-free vaginal moisturising cream cannot only improve vaginal dryness compared to an 0.1% estriol cream but also can relieve dyspareunia as well as improve woman's impairment of daily life, justifying its use as a first choice for mild or moderate vulvovaginal dryness symptoms.

**Data Availability Statement:** All relevant data are within the manuscript and its Supporting Information files.

**Funding:** The funder Dr. August Wolff GmbH & Co. KG Arzneimittel (Bielefeld, Germany) provided support in the form of salaries for authors [SGdA, LG, MH, and CM] and played an additional role in the study design, decision to publish, and preparation of the manuscript. However, the funder had no role in data collection and analysis. The scientific supervisor of the study [PS] was also sponsored by Dr. August Wolff GmbH & Co. KG Arzneimittel. The specific roles of these authors are articulated in the 'author contributions' section. There was no additional external funding received for this study.

**Competing interests:** The authors have read the journal's policy. SGdA, LG, MH, and CM, have the following competing interest: They work for the company Dr. August Wolff GmbH & Co. KG Arzneimittel (Bielefeld, Germany) which is interested in developing products regarding treatment of vulvovaginal atrophy/ vaginal dryness symptoms. The principal investigator of the study [PS] as well as the statistical specialist [TWM] were also sponsored by Dr. August Wolff GmbH & Co. KG Arzneimittel. This does not alter the authors adherence to PLOS ONE policies on sharing data and materials.

**Abbreviations:** AE, Adverse Event; ANCOVA, Analysis of Covariance; AUC, Area Under the Curve; BMI, Body Mass Index; CI, Confidence Interval; Δ, delta; ITT, the Intention To Treat Population; NAMS, The North American Menopause Society; n. s., Not Significant; PP, Per Protocol Population; SD, Standard Deviation; SEM, Standard Error of the Mean; SP, Safety Population; TSS, total severity score; VHI, Vaginal Health Index; VVD, Vulvovaginal Dryness; VVA, Vulvovaginal Atrophy; GSM, Genitourinary Syndrome of the Menopause.

## Introduction

With increasing life expectancy, clinical manifestations of menopausal symptoms are becoming more relevant. Changes in the genitourinary tract associated with decreased estrogenization [1] include decreased vaginal vascularization, lubrication, as well as thinning of the vaginal epithelium [2]. The collagen and water content of the vaginal skin decreases, epithelial cells die and exfoliate leading to a substantial loss of glycogen which is associated with a relevant decrease in the lactobacillus population and an increase in vaginal pH [3]. As a result, epithelium integrity is dry and disturbed [4].

Approximately 50% of postmenopausal women experience vulvovaginal atrophy (VVA) symptoms, also referred to as genitourinary syndrome of menopause (GSM) [4, 5]. The main symptom vulvovaginal dryness negatively affects both, sexual function [6] and overall quality of life [7]. From a subjective perspective, GSM symptoms include dryness, itching, burning sensation and pain (unrelated to sexual intercourse) as well as dyspareunia (pain during sexual intercourse) [2, 5].

According to the NAMS position statement [8], the primary aim of an appropriate GSM treatment is to alleviate the subjective vulvovaginal atrophy symptoms. First-line therapies comprise non-hormonal, long-acting vaginal moisturizers [8, 9]. For symptomatic GSM not responding to hormone-free approaches, low-dose vaginal estriol and vaginal estradiol products are recommended [9, 10]. This statement has been gaining strength in last 10 years even though clinical data comparing the use of hormone-free to hormonal vaginal products are missing.

Several hormone-free vaginal products are available to improve the moisture balance in the vagina as well as to reduce the vulvovaginal symptoms [11]. Many hormone-free formulations are water-based gel formulations lacking lipid components. However, since the vaginal epithelium is a multi-layered stratified squamous epithelium [12], a cream formulation containing both, water and skin-soothing lipids appears to be a promising treatment option [5, 6] to maintain hydration and suppleness [13]. The efficacy of this oil-in-water hormone-free vaginal moisturising cream under investigation, improving subjective vulvovaginal atrophy symptoms and its tolerability have been previously confirmed in pre- and postmenopausal women [14] as well as in breast cancer patients [15].

The aim of this study was to provide evidence regarding efficacy of a vaginal hormone-free moisturising cream compared to a cream with 0.1% estriol.

## Methods

### Study design

This prospective, open-label, randomized, controlled trial (RCT) to show non-inferiority with two parallel groups (hormone-free moisturising cream vs. estriol cream) was conducted at 8 centres in Germany and Switzerland from 07 November 2016 (first enrolment) to 19 June 2017 (last patient finished), according to the Declaration of Helsinki and the GCP guidelines. The study was approved by the Independent Ethics Committee of Schleswig-Holstein (Code121/16II) in Germany and by Kantonale Ethikkommission Bern (Code 2016–01464) in Switzerland, respectively. All women gave written informed consent before starting the study. This clinical trial was registered at EudraCT (2016-002199-28) and at ClinicalTrials.gov (NCT03044652). The study is compliant with CONSORT guidelines for non-inferiority and equivalence randomized trials [16] and with the FDA guideline for non-inferiority clinical trials [17].

## Study population

Only postmenopausal women (amenorrhea > 12 months or bilateral oophorectomy with or without hysterectomy > 3 months) exhibiting subjective vulvovaginal dryness symptoms with a total severity sum score (TSS) ≥ 3, a visual analogue scale (VAS) value "Overall impairment of daily life due to the condition vulvovaginal dryness" > 0, a Papanicolaou test (Pap test) class I or class II and a body mass index (BMI) between 19 kg/m$^2$ and 35 kg/m$^2$ were included in this study.

Women with known allergies to any of the cream ingredients or had used systemic/local hormonal therapies within 3 months before first visit as well as women who used moisturizers (vagina/vulva), systemic corticosteroids, antibiotics or antimycotics 14 days before first visit and/or during the trial were excluded. No violations of the exclusion criteria were detected in the trial.

Patients admitted to the trial were randomized either to receive the estriol cream or to receive the hormone-free moisturising cream in a 1:1 ratio. Separate randomization lists (using randomly permuted blocks with a fixed block size of four) were provided for each centre. Randomization codes were generated centrally by utilizing the software SAS® for Windows version 9.4. Randomization was performed by a statistician who was not directly involved in the analysis of the trial. Upon randomization, each patient received a unique randomization number. The medication and the randomization numbers were allocated to the trial centres according to the randomization lists (S1 File).

## Product application

The reference product (Ovestin® 1 mg creme, Aspen Bad Oldesloe GmbH, Germany), hereafter referred to as "estriol cream" contains 1 mg estriol in 1 g cream and is approved as medicinal product. The investigational product (Vagisan® Moisturising Cream, Dr. August Wolff GmbH & Co. KG Arzneimittel, Germany), is a hormone-free oil-in-water emulsion (cream) and certified as class IIb medical device for intra- and extra-vaginal use (CE 0483). It has a lipid content of 23%, a pH adjusted to 4.5 with lactic acid and an osmolality of 374 mOsm/kg, which is in accordance with the WHO's recommendation of < 380 mOsm/kg for personal lubricants [18, 19].

Both vaginal creams were applied by the patients themselves according to the respective package leaflet. Hormone-free moisturising cream was applied intravaginally, using the provided applicator (approximately 2.5 g) once daily at night, and in the outer genital area (a cream ribbon of approx. 0.5 cm) several times per day, as needed. Once symptoms had improved, the frequency of cream application could be reduced, as needed. The estriol cream (0.5 g) was applied intravaginally once daily for the first 3 weeks with the provided applicator and twice-weekly thereafter, according to package leaflet.

## Study endpoints

The primary objective of this trial was to confirm the non-inferiority of the treatment with hormone-free moisturising cream compared to an estriol cream (0.1%). The comparison was based on the subjective assessment of the symptoms of vulvovaginal dryness (dryness, itching, burning and pain unrelated to sexual intercourse). Symptoms were assessed on Visits 1, 2 and 3 as well as once weekly at home in a patient diary, rating their severity on a five-point scale (0 = none, 1 = mild, 2 = moderate, 3 = severe, 4 = very severe). total severity score was defined as the sum of the single score dryness, itching, burning and pain unrelated to sexual intercourse. In total, the total severity score scale ranges from 0 = no complaints to 16 = very severe

complaints. Differences to Baseline (Visit 1) of the total severity score assessed after 43 days of treatment (Visit 3) served as primary endpoint for the test of non-inferiority.

Secondary endpoints included the participants' assessment of (1) the severity score of single subjective symptoms; (2) the severity scoring of dyspareunia (by sexually active participants only) according to a five-point scale (0 = none, 1 = mild, 2 = moderate, 3 = severe, 4 = very severe), (3) the impairment of daily life due to single subjective symptoms and the impairment of daily life due to dyspareunia assessed by using a VAS ranging from 0 (no impairment) to 10 (very pronounced impairment); (4) the overall impairment of daily life score due to "vaginal dryness", assessed on an VAS which was categorized as follows: 0.1 to 3.3 = mild, 3.4 to 6.6 = moderate, 6.7 to 10 = severe; (5) the Vaginal Health Index (VHI) and the vaginal status of Lactobacillus flora as objective assessment of vaginal findings at each visit by the investigator's clinical evaluation [20]. The Vaginal Health Index is the sum of the scores of the criteria overall elasticity, fluid secretion type and consistency, vaginal pH, epithelial mucosa and moisture which are graded from 1 (worst) to 5 (best). Safety was assessed by evaluating the occurrence of adverse events (AEs).

The severity five-point scale used in this study has already been successfully used in other gynaecological studies [14, 15], while the Vaginal Health Index is a standardized index to assess the health status of the vagina [20]. According to the protocol, patients were asked to report concomitant therapies and adverse events (AEs) on all visits.

## Statistical analysis

172 women, who received at least one application of the test products, regardless of the number of further assessments, were included into the safety population (SP) and 169 of these women with at least one post-baseline assessment were included into the intention to treat population (ITT). 151 women (per protocol population, PP) finished the study in accordance with the trial protocol without major protocol deviations (Fig 1). Study endpoints were analysed primarily for the PP population and repeated, for sensitivity reasons, for the ITT population [16].

According to the study protocol, a one-sided test on differences to baseline of total severity score (Day 43) was carried out in the PP population to determine whether the reduction of this value in the hormone-free moisturising cream group was not clinically inferior compared to the estriol cream group. Delta ($\Delta$) value of 1.5 was defined as the limit for a clinically relevant difference.

A one-sided t-test for independent samples or Mann-Whitney-Wilcoxon test (depending on the result of the Shapiro-Wilk test for normality) was performed. In addition, it was checked whether a two-sided test for $\Delta = 1.5$ and a one-sided test for a ("sharper") $\Delta = 1.0$ also confirmed the non-inferiority of hormone-free moisturising cream.

The sample size was calculated with the Software "N" (IDV, Gauting, Munich) and verified by SAS Sample Size 3.1. A sample size of 138 (69 for each arm) for a one-sided t-test for non-Inferiority ($\alpha = 0.05$) sufficed to assure a statistical power of 90% to reject the null hypothesis. To account for an assumed drop-out rate of 20% a total of n = 172 patients was randomized.

All secondary endpoints were evaluated descriptively and interpreted in the context of an explorative data analysis. The endpoint Area Under the Curve (AUC) over the course of the study was used for the statistical evaluation of the comparison of the two treatments for all secondary endpoints given above and using the t-test or the Mann-Whitney-Wilcoxon test (depending on the Shapiro-Wilk tests for normality).

All results are expressed as the mean and standard deviation (SD) or Standard Error of the Mean (SEM). A p value < 0.05 was considered as statistically significant.

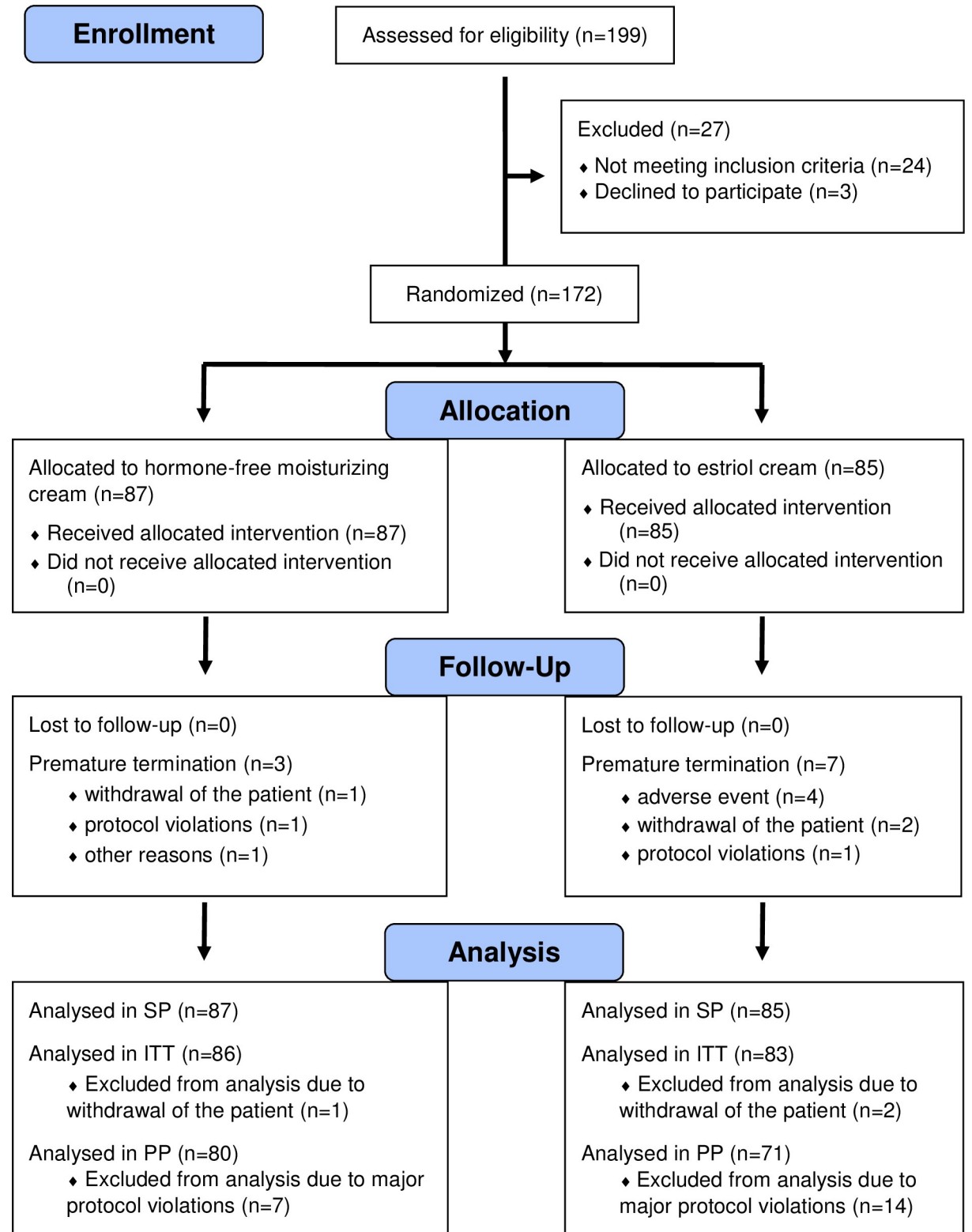

**Fig 1. CONSORT flow diagram.** Disposition of patients. SP, Safety Population; ITT, Intention-To-Treat Population; PP, Per Protocol Population.

# Results

## Study population

In total, 172 women were included in the study, 87 were randomized to the hormone-free moisturising cream group and 85 women to the estriol cream group (Fig 1). 98.8% of women met all eligibility criteria. 21 women were excluded from analysis due to major protocol violations such as antibiotic intake (n = 1) and premature trial termination. Premature termination (n = 10, 5.8%) occurred more than twice as much in the estriol cream group (n = 7, 8.2%) as compared to the hormone-free moisturising cream group (n = 3, 3.4%) (Fig 1). Four of the premature terminations in estriol group were due to AEs such as lower abdominal pain, candida vulvitis, palpitations with rising blood pressure and one broken artificial tooth with surgery. None of the premature termination in the hormone-free moisturising group was related to AEs. Demographic and baseline characteristics are summarized in Table 1.

## Non-inferiority of hormone-free moisturising cream efficacy

The total severity score for both treatments decreased over time (Fig 2). From Day 1 to Day 43, the total severity score for hormone-free moisturising cream decreased from 6.1 ± 0.3 to 1.1 ± 0.2 and for estriol cream group from 6.0 ± 0.3 to 0.6 ± 0.1.

Accordingly, the total severity score decreased by 5.0 (95% confidence interval (CI): 4.6, 5.4) for hormone-free moisturising cream and by 5.4 (95% CI: 4.9, 5.8) for estriol cream group. Analysis of the differences to baseline of the TSS on Day 43 showed that the assumption of the t-test did not hold (normality rejected by Shapiro-Wilk test). Wilcoxon-Mann-Whitney Wilcoxon was used instead. The result was significant (p = 0.0002) supporting the non-inferiority of hormone-free in comparison to estriol. Therefore, the analysis of the differences to baseline of the total severity score on Day 43 (one-sided test; $\Delta = 1.5$) confirmed hormone-free moisturising cream to be noninferior as compared to the estriol cream (PP: p < 0.001; ITT: p < 0.001). Non-inferiority could also be verified with two-sided testing ($\Delta = 1.5$) and even with a "sharper" one-sided testing ($\Delta = 1.0$).

After 1 week of treatment (Day 7), total severity score decreased 49% and 48% by hormone-free moisturising cream and estriol cream, respectively (Fig 2). Symptom relief improved over

**Table 1. Demographic and baseline characteristics.**

| Characteristics | Values per treatment group | |
|---|---|---|
| | **Hormone-free cream** | **Estriol cream** |
| **Number of Women PP (ITT)** | 80 (86) | 71 (83) |
| **Age (years)** | 59.5 ± 7.3 | 61.7 ± 6.9 |
| **BMI (kg/m²)** | 25.8 ± 3.8 | 25.9 ± 3.7 |
| **Sexually active (n)** | 64 | 42 |
| **Severity Scoring of individual symptoms at baseline*** | | |
| **Dryness** | 2.3 ± 0.8 | 2.3 ± 0.9 |
| **Itching** | 1.4 ± 0.1 | 1.4 ± 0.1 |
| **Burning** | 1.4 ± 0.1 | 1.5 ± 0.1 |
| **Pain unrelated to sexual intercourse** | 1.1 ± 0.9 | 1.1 ± 1.0 |
| **Dyspareunia (if sexually active)** | 2.6 ± 0.1 | 2.7 ± 0.2 |

Data are given as Mean ± standard desviation.

*Severity scale of individual subjective symptoms were assessed using a five-point scale 0 = none; 1 = mild; 2 = moderate; 3 = severe; 4 = very severe.

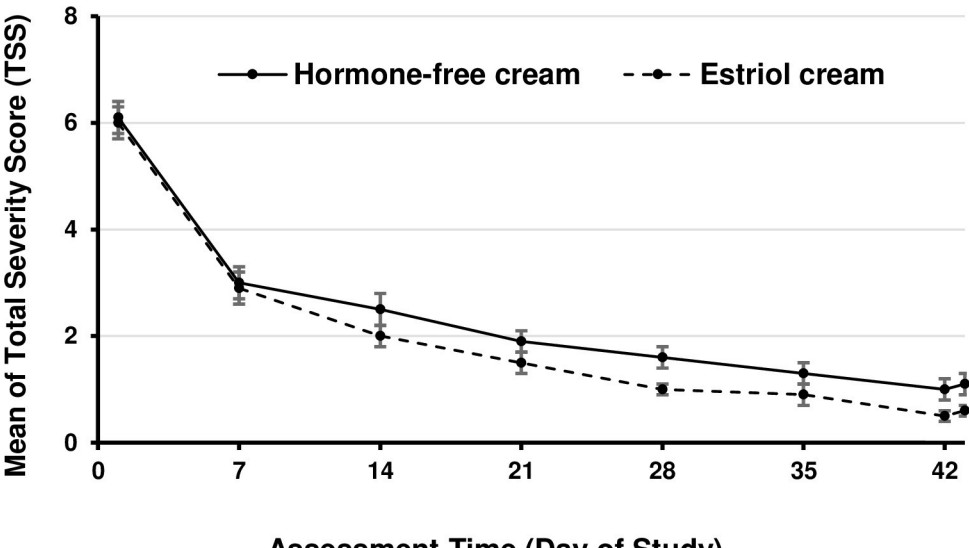

**Fig 2. Mean of Total Severity Score (TSS) over time.** Mean of TSS (sum score of the single subjective symptom parameters: dryness, itching, burning and pain unrelated to sexual intercourse) over time until Day 43. Scale of TSS: 0 = no complaints to 16 = very severe complaints. Values represent Mean ± SEM. Number of patients (PP): Hormone-free moisturizing cream n = 80; Estriol cream n = 71.

time in a similar manner for both vaginal creams. Thus, the relief of symptoms was achieved in a similar time frame for both creams (Fig 2).

## Efficacy assessment of subjective symptoms

The mean severity score for each individual subjective symptom significantly decreased over time in both treatment groups (data not shown). At baseline, most of the women suffered from at least mild vulvovaginal subjective symptoms in both treatment groups (Fig 3).

Dryness was the symptom that affected more than 96% of the women in both treatment groups. The percentage of symptom-free women increased progressively with the duration of the treatment in both studies (Fig 3). At the end of the hormone-free moisturising cream treatment, 63.8% of the women reported to be symptom-free with respect to dryness, 78.8% with respect to itching and 77.5% to burning as well as 87.5% to pain unrelated to sexual intercourse (for estriol cream 76.1%, 80.3%, 91.5%, and 94.4%, respectively). None of the women in any treatment group evaluated their symptoms as severe or very severe on Day 43. An improvement of the symptom dryness until Day 43 was reported by 91.3% of women in the hormone-free moisturising cream group and by 95.8% of women in the estriol cream group, respectively (Fig 4). An improvement of the symptoms itching and burning was reported by 75.0% and 73.8% of the women in of the hormone-free moisturising cream group and by 80.3% and 85.9% in the estriol cream group, respectively (Fig 4).

## Efficacy assessment of dyspareunia severity (if sexually active)

More than 95% of the sexually active women (n = 106) were affected by dyspareunia at Day 1. From Day 1 to Day 43, the mean severity score for dyspareunia improved by 1.7 (95% CI: 2.0, 1.3) for the hormone-free moisturising cream and by 2.2 (95% CI: 2.6, 1.8) for estriol cream, respectively. Comparing the AUC of the severity score for dyspareunia among sexually active women, no statistically significant difference between both treatments was found (p = 0.398).

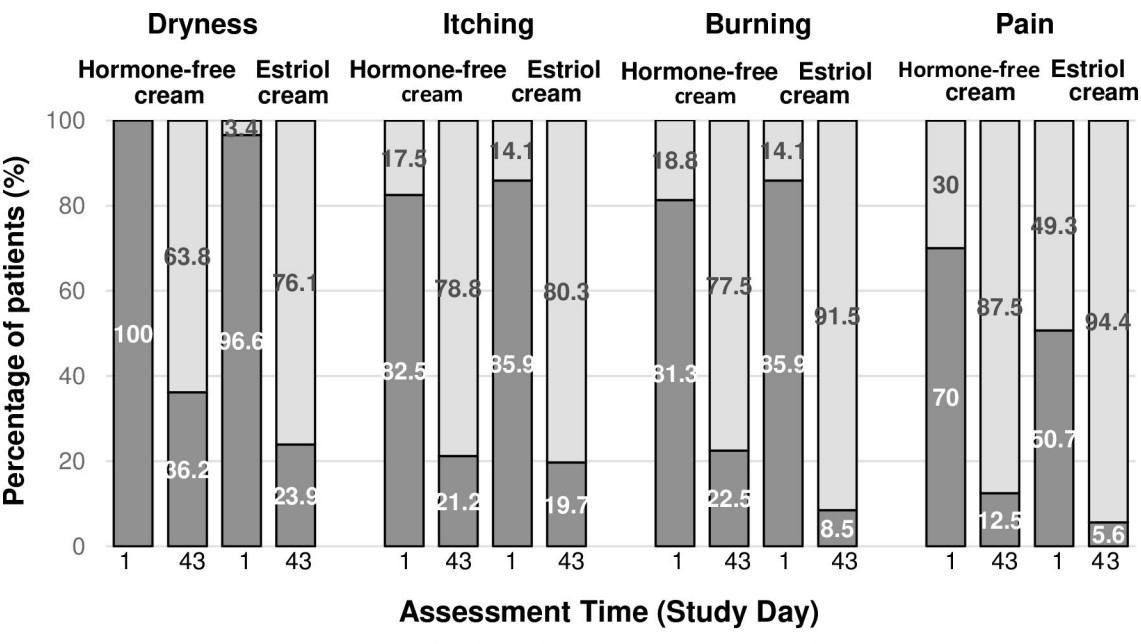

**Fig 3. Vulvovaginal subjective symptoms at baseline.** Percentage of patients suffering from at least mild subjective symptoms or free of symptoms. Number of patients (PP): Hormone-free moisturizing cream n = 80; Estriol cream n = 71.

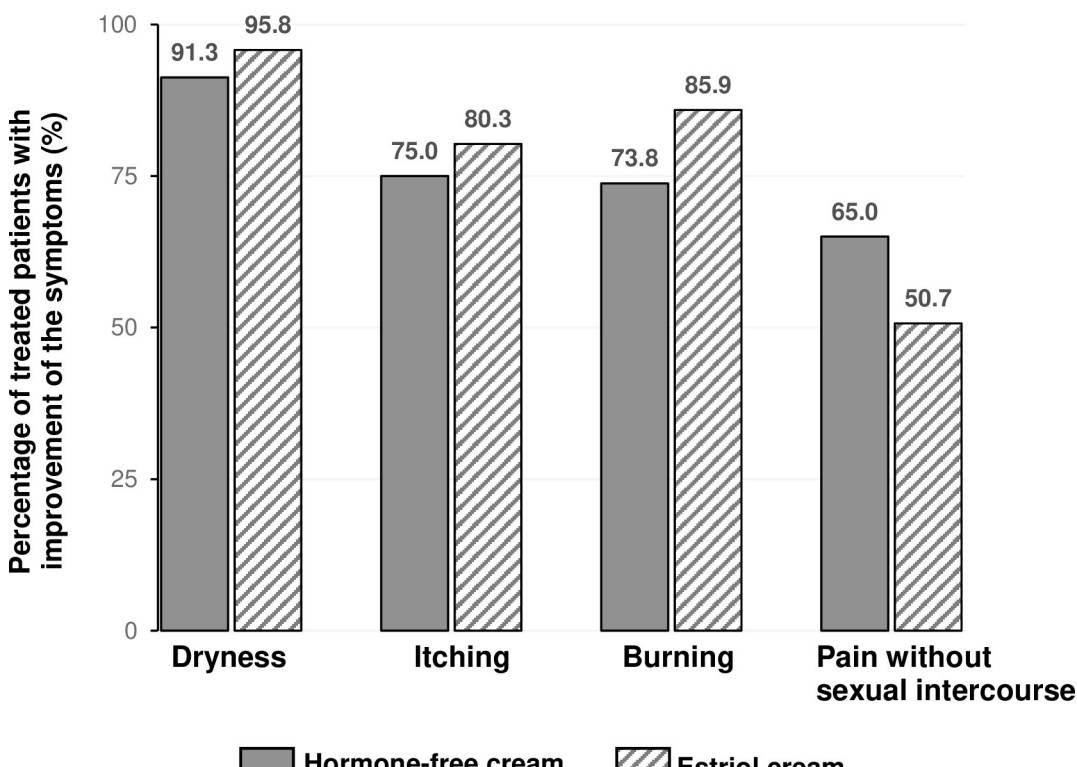

**Fig 4. Vulvovaginal subjective symptoms at the end of the treatment.** Percentage of treated patients with improvement of their individual subjective symptoms at the end of the treatment (Day 43). Number of patients (PP): Hormone-free moisturizing cream n = 80; Estriol cream n = 71.

At Day 1, 59.4% and 41.5% of sexually active women rated dyspareunia as severe or very severe in the hormone-free moisturising cream and estriol cream groups, respectively (Fig 5). On Day 43 of treatment an improvement of the symptom dyspareunia was reported by 84% of the hormone-free moisturising cream treated women and 87% of the estriol cream treated women. In the hormone-free cream group, 38.6% of the women reported to be symptom-free and 57.9% in estriol cream group, respectively.

## Overall impairment of daily life due to individual subjective symptoms

Both treatments improved the impairment of daily life due to each individual subjective symptom over time as compared to baseline (data not shown). Interestingly, when comparing both treatment groups, no significant difference was detected for each individual subjective symptom after 43 days of treatment (Fig 6). An improvement of the impairment of daily life (Day 43) due to individual subjective symptoms was reported by 96% of the hormone-free moisturising cream treated women and by 99% by of the estriol cream treated women.

## Impairment of daily life due to the condition dyspareunia (if sexually active)

The impairment of daily life due to dyspareunia for sexually active women improved in both treatment groups over time. In the hormone-free moisturising cream group, the mean score

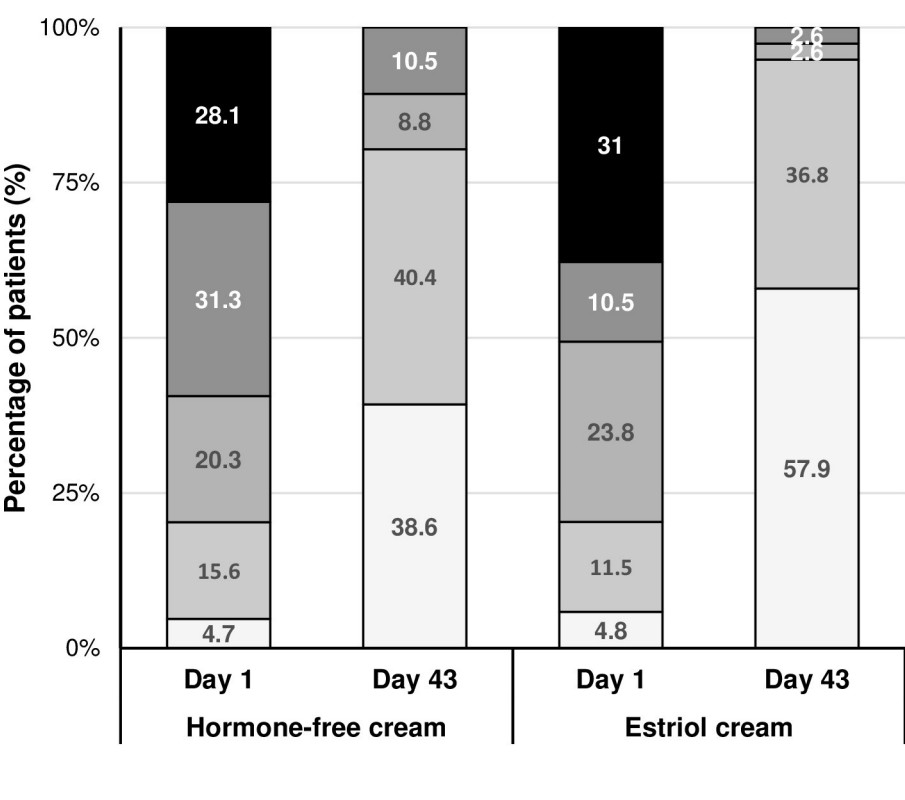

**Fig 5. Dyspareunia severity.** Percentage of patients for the assessment of dyspareunia severity (if sexually active). The total number of patients refers to sexually active patients at Day 1/ Day 43: Hormone-free moisturizing cream n = 64/ 57; Estriol cream n = 42/37. Scale severity score for dyspareunia: 0 = none, 1 = mild, 2 = moderate, 3 = severe, 4 = very severe.

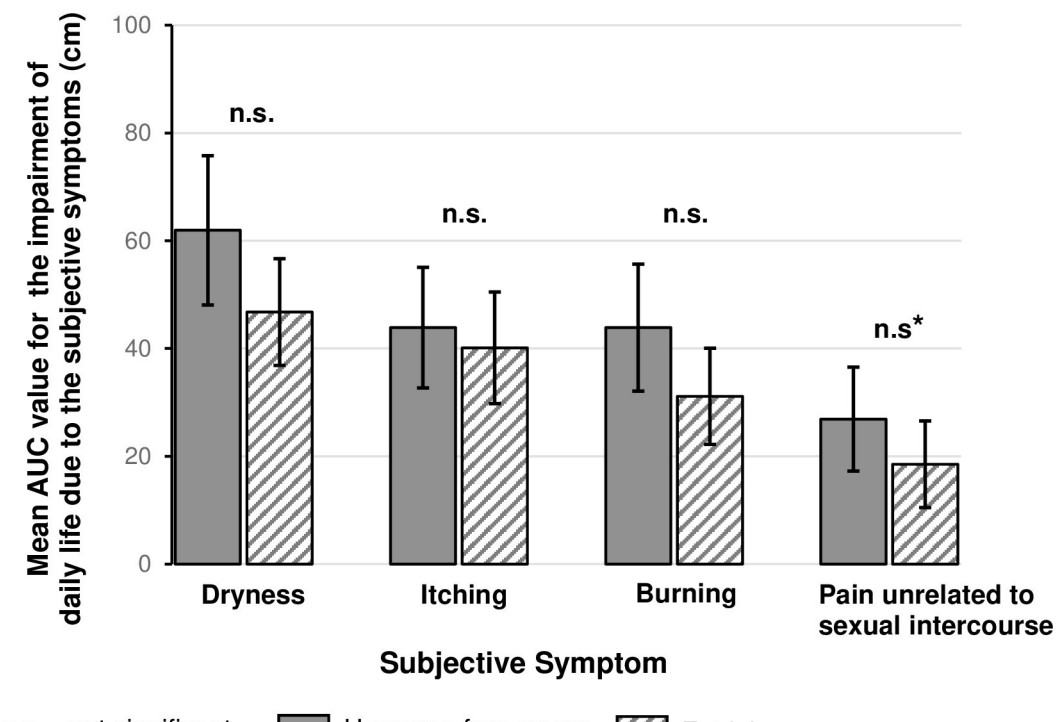

**Fig 6. Comparison of the impairment of daily life due to each of the subjective symptoms (AUC).** Values represent Mean ± 95% CI. Not significant (n.s.). *Due to different baseline values, an ANCOVA using the baseline value as covariate was performed. The result demonstrated no significant difference at Day 43 between treatments.

decreased from 7.03 ± 0.34 to 1.56 ± 0.32. In the estriol cream group, the mean dyspareunia score decreased from 6.60 ± 0.45 to 0.65 ± 0.27. Comparison of AUC values of both treatment groups revealed no statistically significant differences (p = 0.066) for the improvement of the impairment of daily life due to the condition dyspareunia. An improvement of the impairment of daily life due to dyspareunia until Day 43 was reported by 98.2% of the women in the hormone-free moisturising cream group and by 86.8% of the estriol-treated women.

## Overall impairment of daily life due to the condition "vaginal dryness"

The overall impairment of daily life due to the condition "vaginal dryness" for both treatments was statistically significantly different in favour of the estriol cream group (p < 0.01). A sub-group analysis stratified by the severity of the impairment of daily life due to the condition "vaginal dryness" showed that the difference between both groups was only significant for women who started the study with a severe overall impairment (p < 0.01) but not for those with mild (p = 0.298) or moderate (p = 0.086) impairment (Fig 7).

## Objective assessment of vaginal findings (VHI)

On Day 1, the overall elasticity was evaluated as poor for up to 40% of the women in both treatment groups. The mean difference to baseline on Day 43 was 1.0 ± 0.1 for the hormone-free moisturising cream-treatment group and 1.8 ± 0.1 for the estriol-treatment group. An improvement of the overall elasticity until Day 43 was reported by 67.5% of the hormone-free moisturising cream-treated women and 84.5% of the estriol treated women.

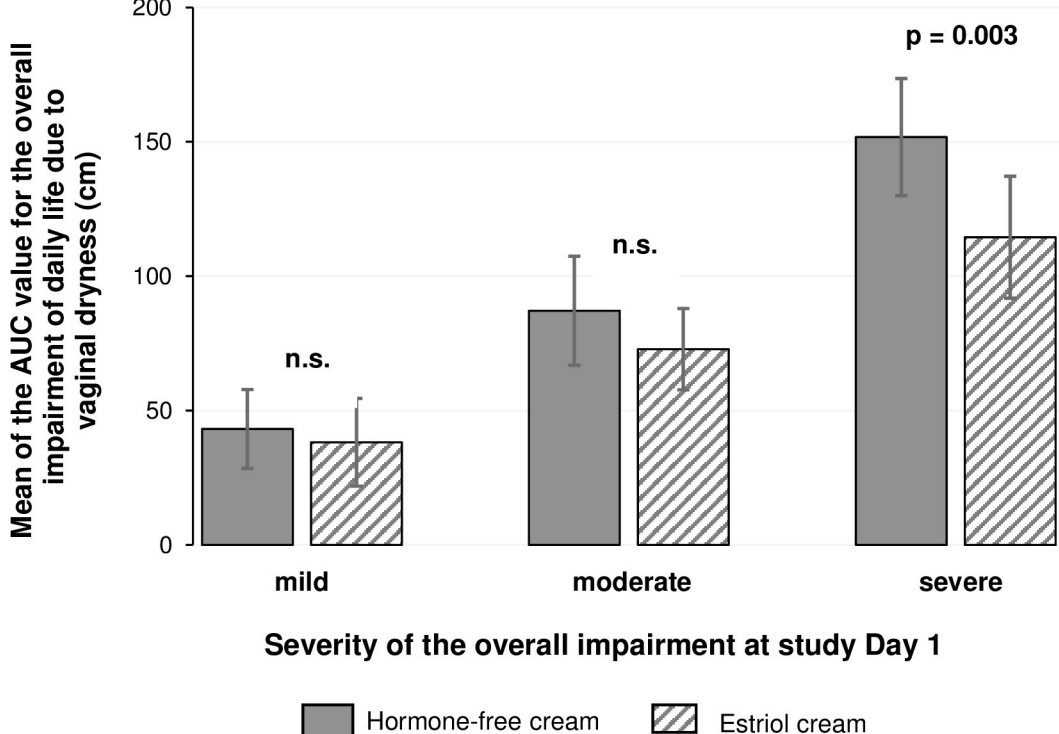

**Fig 7. Overall impairment of daily life due to the condition "vaginal dryness".** Number of patients (PP) with mild/
moderate/severe "vaginal dryness" in each treatment group: Hormone-free moisturizing cream n = 20/24/32; Estriol cream
n = 19/30/21. Mean ± 95% CI of the AUC value. Not significant, p > 0.05.

On Day 1, the fluid secretion was evaluated as scant and thin yellow for up to one-third of
the women in both treatment groups. The mean difference to baseline on Day 43 was 1.2 ± 0.1
for the hormone-free moisturising cream-treatment group and 1.9 ± 0.1 for the estriol-treat-
ment group. An improvement of the fluid secretion until Day 43 was reported by 73.8% of the
hormone-free moisturising cream-treated women and 85.9% of the estriol-treated women.

On Day 1, the vaginal pH was evaluated as greater or equal to 6.1 for more than one-third
of the women in both treatment groups. The mean difference to baseline on Day 43 was
0.3 ± 0.2 for the hormone-free moisturising cream-treatment group and 1.9 ± 0.2 for the
estriol-treatment group. An improvement of the vaginal pH until Day 43 was reported by
41.3% of the treated women and 73.2% of the estriol-treated women.

On Day 1, petechiae noted before contact was documented for none of the women of the
hormone-free moisturising cream-treatment group and for 4.2% of women of the estriol-treat-
ment group (n = 3). The mean difference to baseline on Day 43 was 0.4 ± 0.1 for the group
treated with hormone-free moisturising cream and 1.0 ± 0.1 for the estriol-treatment group,
respectively. An improvement of the epithelial mucosa until Day 43 was reported by 36.3% of
the hormone-free moisturising cream-treated women and 81.7% of the estriol-treated women.

Both study treatments significantly improved the mean Vaginal Health Index score com-
pared to baseline (p < 0.001). Comparison between the treatments (AUC values) revealed a
statistically significant difference (p < 0.001) for all criteria of the Vaginal Health Index (elas-
ticity, fluid secretion, vaginal pH, moisture, and epithelia mucosa) in favour of the estriol
cream (Fig 8). Nearly 80% of the hormone-free moisturising cream treated women and more
than 95% of the estriol cream treated women showed improvement of Vaginal Health Index

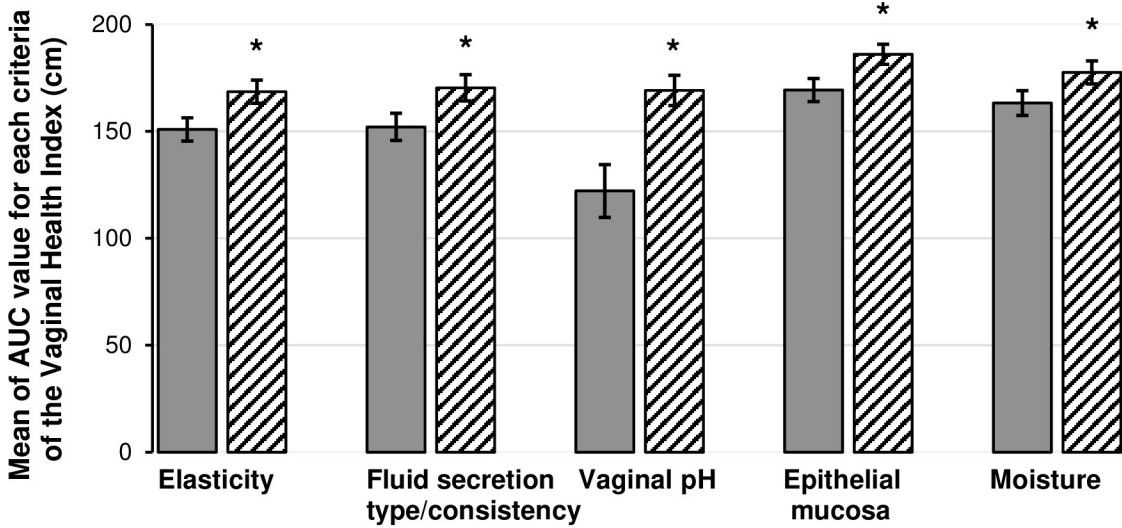

**Fig 8. Objective assessment of vaginal findings.** Improvement of all Vaginal Health Index criteria (objective assessment of vaginal findings) and significant differences in favor of the estriol cream. Number of patients (PP): Hormone-free moisturizing cream n = 74; Estriol cream n = 69. Mean AUC values ± 95% CI. Mann-Whitney-Wilcoxon tests were applied to evaluate any differences in the effect of the treatments (statistically significant, p < 0.001).

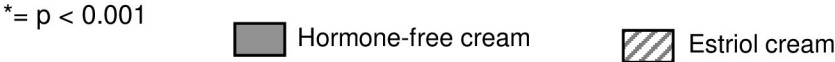

over time. An increase of the percentage of women with a Lactobacillus status within the normal range was, as expected, only observed in the estriol cream group (47.9%).

## Global judgement of efficacy and tolerability

The efficacy was judged good or very good by more than 80% of the women treated with the hormone-free moisturising cream and by more than 90% of the estriol-treated women. Tolerability of both creams was judged good or very good by 94% and 95% of the physicians, respectively.

## Safety assessment

A total of 192 non-serious adverse events was reported in this study by 79 women: 33 (37.9%) out of 87 women in the hormone-free moisturising cream group and 46 (54.1%) out of 85 women in the estriol cream group. Most of adverse events were mild (71.9%) or moderate (26.0%). Only 4 (2.1%) severe adverse events occurred: 2 unrelated to treatment, 1 probably (headache) and 1 certainly (local burning) related to the estriol cream. adverse events considered serious did not occurred in this trial. The low number of non-serious adverse events related to the hormone-free moisturising cream gave no indication for safety concerns in this study. Overall, more adverse events with a certain, probable or possible relation to the treatment occurred in the estriol group.

## Discussion

The total severity score, as primary endpoint confirmed the non-inferiority of the hormone-free moisturising cream as compared to an estriol cream regarding the relief of subjective

symptoms (dryness, itching, burning and pain unrelated to sexual intercourse) in postmeno-pausal women. Accordingly, the efficacy difference between both treatments was not clinically relevant. Importantly, feeling of dryness, the most frequent symptom at baseline improved in more than 90% of women in both treatment groups. Dyspareunia was also significantly reduced during the study without showing significant differences between both treatments. The relief of the subjective symptoms and dyspareunia over time correlates with an improve-ment of women's impairment of daily life for both vaginal treatments.

A significant difference between hormone-free moisturising cream treatment and estriol cream treatment in favour of the group receiving the estriol cream was detected only in a sub-group of women who started the study with severe overall impairment of daily life due to the condition "vaginal dryness". Accordingly, these women would be advised to use a vaginal estriol cream from the beginning, if they wish. On the other hand, the results confirm that women with mild to moderate impairment of daily life due to the condition "vaginal dryness" benefit from both creams. In addition, the hormone-free moisturising cream significantly improved objective signs assessed by measurement of the Vaginal Health Index as compared to baseline. However, this improvement was significantly more pronounced after treatment with the estriol cream. Nearly 80% of the hormone-free moisturising cream-treated women and more than 95% of the estriol-treated women showed an improvement of the Vaginal Health Index over time.

The therapeutic effects of this oil-in water cream are related to the properties of the formu-lation, primarily the high-water content and the lipids supplied with the cream. The use of this moisturising cream is a purely symptomatic therapy where the skin of the vagina and on the vulva is provided with both, moisture and soothing lipids, similarly as for emollients used in the treatment of atopic dermatitis.

This is the first time that this hormone-free moisturising cream has been compared with an estriol cream with regard to improvement of both, subjective symptoms and objective signs of vulvovaginal atrophy over a total study duration of 6 weeks. In a previous study, this oil-in-water hormone-free vaginal moisturising cream showed clinical superiority in reducing sub-jective vulvovaginal dryness symptoms compared to a hormone-free moisturising gel contain-ing hyaluronic acid in pre- and post-menopausal women [14].

The clinically relevant efficacy of vaginal estriol regarding the relief of subjective symptoms and objective findings compared to placebo or other vaginal estrogen preparations, has been sustained in several studies [21–23], reviews [24, 25] and meta-analyses [26]. In addition, the reference estriol product chosen in this study has been approved for the therapy of vulvovagi-nal atrophy symptoms due to estrogen deficiency for more than 25 years in Germany.

Consequently, the choice of an estriol cream (0.1%) as a reference product for clinically effective treatment of vulvovaginal atrophy symptoms corroborates the efficacy of the hor-mone-free vaginal cream under investigation which has been shown in this study to be non-inferior with regard to the improvement of subjective symptoms of vulvovaginal atrophy. These results provide convincing evidence to recommend the use of this hormone-free mois-turising vaginal cream as first-line therapy to relieve vulvovaginal atrophy-related symptoms, as recommended by NAMS (2020) [8].

This new information expands knowledge about the benefits of using this hormone-free mois-turising cream and supports the recommendation that non-hormonal treatment options are rele-vant for all women who wish to reduce or avoid local hormone treatment, if medically advised.

### Limitations

The hormone-free moisturising cream under investigation does not contain any pharmacolog-ically active moieties. Due to the kind of packaging of the test products (different size of tube

and applicator) and the different frequency of application (dosage) a blinded study design was not feasible. Furthermore, more sexually active women were randomized to the hormone-free cream group (n = 64) as compared to the estriol cream group (n = 42). The parameter "sexually active" was not part of the inclusion criteria. After randomisation, differences in the number of "sexually active" participants between the groups were detected but did not affect the primary outcome of the study. The delay in publication of this study was caused by internal reasons.

## Conclusion

The efficacy of the hormone-free moisturising vaginal cream, which is non-inferior to that of the estriol cream, the very good tolerability, the low number of adverse events as well as the satisfaction of the treated women illustrate that the hormone-free vaginal cream is a reliable therapy option for the treatment of vulvovaginal atrophy symptoms in postmenopausal women, especially in women for whom hormone therapy is contraindicated or undesirable.

## Supporting information

**S1 File. Clinical trial protocol–Synopsis.**
(PDF)

**S1 Checklist. CONSORT 2010 checklist of information to include when reporting a randomised trial**\*.
(PDF)

## Acknowledgments

The authors are grateful for the support of the coordinating investigator and the participating centers. We thank proDERM (Hamburg, Germany) for their participation in data collection, analysis, and interpretation of data.

## Author Contributions

**Conceptualization:** Clarissa Masur, Petra Stute.

**Formal analysis:** Theodor W. May.

**Funding acquisition:** Clarissa Masur.

**Methodology:** Theodor W. May.

**Supervision:** Manuel Häuser, Petra Stute.

**Visualization:** Susana Garcia de Arriba.

**Writing – original draft:** Susana Garcia de Arriba, Lisa Grüntkemeier.

**Writing – review & editing:** Susana Garcia de Arriba, Lisa Grüntkemeier, Manuel Häuser, Clarissa Masur, Petra Stute.

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
