## [Decision Letter · Decision Letter 0]

11 Feb 2022

PONE-D-21-39841Vaginal hormone-free moisturising cream is not inferior to an estriol cream for treating symptoms of vulvovaginal atrophy: prospective, randomised studyPLOS ONE

Dear Dr. Garcia de Arriba,

Thank you for submitting your manuscript to PLOS ONE. After careful consideration, we feel that it has merit but does not fully meet PLOS ONE’s publication criteria as it currently stands. Therefore, we invite you to submit a revised version of the manuscript that addresses the points raised during the review process.

We look forward to receiving your revised manuscript.

Kind regards,

Walid Kamal Abdelbasset, Ph.D.

Academic Editor

PLOS ONE

Journal Requirements:

(This work was supported by The Research Fund of Rigshospitalet, Copenhagen University Hospital [grant number E-22515-01] awarded to APM and ØL. URL: https://www.forskningspuljer-rh.dk/

Ole Kirks Foundation [no grant number] awarded to HSN. URL: https://www.olekirksfond.dk/)

(I have read the journal's policy and the authors of this manuscript have the following competing interests: The authors report no conflicts of interest for the submitted research. HSN reports grants from Freya Biosciences ApS, Ferring Pharmaceuticals, BioInnovation Institute, Ministry of Education, Novo Nordisk Foundation, Augustinus Foundation, Oda and Hans Svenningsens Foundation and honoraria from Ferring Pharmaceuticals, Merck A/S, Astra Zenica, Cook Medical, and DW reports a grant from Novo Nordisk Foundation during the conduct of this study.)

We note that you received funding from a commercial source: (Freya Biosciences ApS, Ferring Pharmaceuticals, BioInnovation Institute, Ministry of Education, Novo Nordisk Foundation, Augustinus Foundation, Oda and Hans Svenningsens Foundation, Merck A/S, Astra Zenica, Cook Medical,Novo Nordisk)

(Competing interests: The authors have read the journal’s policy. SGdA, LG, MH, and CM, have the following competing interest: They work for the company Dr. August Wolff GmbH & Co. KG Arzneimittel (Bielefeld, Germany) which is interested in developing products regarding treatment of vulvovaginal atrophy/ vaginal dryness symptoms. The principal investigator of the study [PS] as well as the stadistical specialist [TWM] were also sponsored by Dr. August Wolff GmbH & Co. KG Arzneimittel. This does not alter the authors adherence to PLOS ONE policies on sharing data and materials.)

We note that one or more of the authors are employed by a commercial company: name of commercial company. 

5. Please include your tables as part of your main manuscript and remove the individual files. Please note that supplementary tables (should remain/ be uploaded) as separate "supporting information" files

Reviewers' comments:

Reviewer's Responses to Questions

**Comments to the Author**

1. Is the manuscript technically sound, and do the data support the conclusions?

Reviewer #1: Yes

Reviewer #2: Yes

Reviewer #3: Yes

2. Has the statistical analysis been performed appropriately and rigorously? 

Reviewer #1: No

Reviewer #2: Yes

Reviewer #3: Yes

3. Have the authors made all data underlying the findings in their manuscript fully available?

Reviewer #1: Yes

Reviewer #2: Yes

Reviewer #3: No

4. Is the manuscript presented in an intelligible fashion and written in standard English?

Reviewer #1: Yes

Reviewer #2: Yes

Reviewer #3: Yes

5. Review Comments to the Author

Reviewer #1: This is a noninferiority study of a vaginal cream with an ordinal outcome collected longitudinally. My main concerns are:

1. "TSS was defined as the sum of the single score for all subjective symptoms at each time point." It is not clear what this statement means, but I THINK it means that each 4-point score was evaluated at 4 time points, and they were added? That makes no sense to me. A longitudinal analysis would look at the rate of change of the scores. Alternatively, a primary outcome assessed in a non-longitudinal way could look at the change from baseline at end of study. In any event, I don't know what summing scores over time points means. Or maybe I am interpreting your sentence incorrectly.

2. Sample size is based on a t-test, and there is no indication that your 4-point scores are normally distributed, even when added. I think this calls for sample size determined for the Wilcoxon test. Whether there is software for SS in noninferiority for nonparametric tests I am not sure, but the statistician should be able to figure it out by assuming a proversion parameter.

3. The conclusions should clearly state whether the assumptions underlying the sample size computation were realized in the clinical trial.

Minor point: Please give the block size when using permuted block. Is there any concern about selection bias with a fixed block size?

Reviewer #2: Dear authors,

Did you have performed a sample size calculation? you need to mention that as it was a clinical trial and prospective.

did you evaluate and collect the primary and secondary endpoints of the study during the treatment period? or did you assess the clinical improvements after the treatment period and if yes, how long was it after the treatment?

I think it is quite relevant to make this point clear enough. Also, the time scale of the improvement was not clear. was the estriol associated with longer term improvement than the emulsion?

Moreover, the pattern of application of the emulsion versus the estriol was very different: the emulsion was applied externally several times as needed, while the estriol cream was applied intravaginally once daily for the first 3 weeks and twice-weekly thereafter.

did the frequent application of the emulsion give a false perception of improvement? It might have contributed (bias) to the final results.

can you please elaborate enough and make these essential issues clear?

Reviewer #3: The research has been written in an acceptable manner.

Please just add full names for all abbreviations then proceeded on .

All the sections were recorded in a suitable way. No need for further assessments

6. PLOS authors have the option to publish the peer review history of their article (what does this mean?). If published, this will include your full peer review and any attached files.

Reviewer #1: No

Reviewer #2: **Yes: **Naser Al-Husban

Reviewer #3: No

---

## [Author Response · Author response to Decision Letter 0]

11 Mar 2022

Point-to-Point-Reply

We are grateful to the Reviewers and the Editor for their careful revision and the overall positive feedback we received. Accordingly, we took into account the suggestions of the Reviewers as follows:

In reply to Reviewer 1

1.1 "TSS was defined as the sum of the single score for all subjective symptoms at each time point." It is not clear what this statement means, but I THINK it means that each 4-point score was evaluated at 4 time points, and they were added? That makes no sense to me. A longitudinal analysis would look at the rate of change of the scores. Alternatively, a primary outcome assessed in a non-longitudinal way could look at the change from baseline at end of study. In any event, I don't know what summing scores over time points means. Or maybe I am interpreting your sentence incorrectly.

We thank the Reviewer for this valuable comment. Indeed, the wording creates confusion, so we have modified the sentence to read as follows: TSS was defined as the sum of the single score dryness, itching, burning and pain unrelated to sexual intercourse (see Page 7).

1.2. Sample size is based on a t-test, and there is no indication that your 4-point scores are normally distributed, even when added. I think this calls for sample size determined for the Wilcoxon test. Whether there is software for SS in noninferiority for nonparametric tests I am not sure, but the statistician should be able to figure it out by assuming a proversion parameter.

The primary goal of a sample size calculation is to ensure that the number of subjects in a clinical trial is large enough to provide a reliable answer to the questions addressed (cp. ICH E9 Statistical principles for clinical trials). The question of our study was whether the vaginal hormone-free moisturising cream is not inferior to an estriol cream for treating symptoms of vulvovaginal atrophy. The results of the Wilcoxon-Mann-Whitney test provided enough evidence in favour of the non-inferiority of hormone-free in comparison to estriol (p=0.0002), i.e. the sample size was large enough to provide a reliable answer. 

According to the statistical analysis plan (SAP), the Wilcoxon-Mann-Whitney test (and not the t-Test) was performed because the Shapiro-Wilk test rejected the hypothesis that the differences to baseline of the Total Severity Score (TSS) at 6 weeks for patients treated with the test product and the active control were normally distributed. In this context, we would like to clarify that the assumption of normal distribution was not related to the sum of the 4-point scores (TSS), but the differences to baseline of the Total Severity Score (TSS) at 6 weeks. When we planned this study, we have no reliable information on the distribution of these differences and, especially, no reliable information that these differences are not normally distributed. It should be noted that the two-sample t-Test is comparatively robust under violations of the normality distribution, particularly when the sample sizes are equal (e.g. Zimmerman DW. J Exp Educat. 1987;55(3):171-174; Heeren T, D'Agostino R. Stat Med. 1987;6(1):79-90.). Equal sample sizes were planned for the sample size calculation in this study. 

We agree with the reviewer that based on the meanwhile available study data; a Wilcoxon-Mann-Whitney test would have been more appropriate for a sample size calculation. However, it should be noted that sample size calculations for the Wilcoxon-Mann-Whitney -test are by no means an easy task. There are many different proposed procedures for sample size planning for the Wilcoxon-Mann-Whitney test (e.g. Happ et al. Stat Med. Feb 10;38(3):363-375; Zhu X. Int J Clin Trials. 2021 Aug;8(3):184-195). Most methods and software for sample size calculation for the Wilcoxon-Mann-Whitney test (e.g. R package “samplesize”, GPower) assume very specific models or types of data to simplify calculations ordered categorical or metric data, location shift alternatives, etc. 

In summary, sample size calculations are always based on assumptions. Whether these assumptions are actually met or not, can often only be evaluated when the study has been completed.

We have modified the sentence to read as follows: Analysis of the differences to baseline of the TSS on Day 43 showed that the assumption of the t-test did not hold (normality rejected by Shapiro-Wilk test). Wilcoxon-Mann-Whitney test was used instead. The result was significant (p=0.0002) supporting the non-inferiority of hormone-free in comparison to estriol. (Page 10-11)

1.3. The conclusions should clearly state whether the assumptions underlying the sample size computation were realized in the clinical trial.

The assumption that the underlying differences to baseline of the Total Severity Score (TSS) at 6 weeks for patients treated with the hormone-free cream and the estriol cream (active control) were normally distributed was not correct. Thus, t-test was not the appropriate test for the calculation.as the reviewer correctly mentions Wilcoxon-Mann-Whitney test was used instead. The results of the Wilcoxon-Mann-Whitney Test provided enough evidence in favour of the non-inferiority of hormone-free in comparison to estriol (p=0.0002), i.e. the sample size was large enough to provide a reliable answer. Moreover, the calculated sample size of n=172 patients included assumed a drop-out rate of 20%. However, only 10 women (5.8%) terminated early from the trial after they applied their product cream at least once.

1.4. Minor point: Please give the block size when using permuted block. Is there any concern about selection bias with a fixed block size?

As suggested by the Reviewer, we added the information concerning the block size, which was four, in the manuscript (see Page 6). We have no concerns about selection bias with a fixed block size.

In reply to Reviewer 2

2.1 Did you have performed a sample size calculation? you need to mention that as it was a clinical trial and prospective

We thank the Reviewer for this valuable comment. Yes, a sample size calculation was performed, and you can find this information on the section Statistical Analysis (see Page 8). Please notice that the study title contains also the following information “prospective, randomised study”. 

2.2 Did you evaluate and collect the primary and secondary endpoints of the study during the treatment period? or did you assess the clinical improvements after the treatment period and if yes, how long was it after the treatment? I think it is quite relevant to make this point clear enough.

Indeed, primary and secondary endpoints were collected during the treatment period up to Visit 3 (Day 43). No Follow-up Visit was performed to evaluate efficacy or safety parameters after end of treatment.

2.3 Also, the time scale of the improvement was not clear. 

a) was the estriol associated with longer term improvement than the emulsion?

b) Moreover, the pattern of application of the emulsion versus the estriol was very different: the emulsion was applied externally several times as needed, while the estriol cream was applied intravaginally once daily for the first 3 weeks and twice-weekly thereafter.

c) Did the frequent application of the emulsion give a false perception of improvement? 

It might have contributed (bias) to the final results.

can you please elaborate enough and make these essential issues clear?

We thank the Reviewer for this comment. 

a) This study did not investigate the duration of the treatment effects. Therefore, it could not compare the long-term effects of both vaginal creams. 

b) Further, the timing (intravaginal application in the evening), frequency and dose was chosen according to the recommendations given for both marketed products as described in the respective package inserts. Thus, Hormone-free moisturising cream was applied intravaginally, using the provided applicator (approximately 2.5 g) once daily at night, and in the outer genital area (a cream ribbon of approx. 0.5 cm) several times per day, as needed. Once symptoms had improved, the frequency of cream application could be reduced, as needed. The estriol cream (0.5 g) was applied intravaginally once daily for the first 3 weeks with the provided applicator and twice-weekly thereafter, according to package leaflet.

c) After about a week of daily intravaginal application, the patient applies the Hormone-free cream much less frequently. Also, the external application of the hormone-free cream was applied just as patient´s needs and affects the external genital area. However, it has no effect on the intravaginal area which is responsible for the symptoms that were evaluated in the clinical trial.

In reply to Reviewer 3 

The research has been written in an acceptable manner.

Please just add full names for all abbreviations then proceeded on.

All the sections were recorded in a suitable way. No need for further assessments

We thank the Reviewer for the positive comments. Concerns to be addressed: None, except that the Reviewer requested minor changes. As recommended, we have added full names for all abbreviations within the manuscript.

In reply to the Editor

1. We made the required changes in the manuscript in order to meet PLOS ONE's style requirements. Changes are labelled “Track Changes” in the manuscript (marked-up copy)

2. Figures have been cited as “Fig”.

3. A table with caption as well as all Figure captions have been placed in the manuscript text in read order, immediately following the paragraph where the Table/figure is first cited.

4. Text within Figures were corrected to Arial (8-12 point).

5. All Figure files format was changed to TIFF format using PACE tool. New figure files.

6. Abbreviations have been revised.

7. Updated Funding statement (see below)

8. Name of commercial company: Dr. August Wolff GmbH & Co. KG Arzneimittel (Bielefeld, Germany)

9. Amended Competing Interests Statement (see below)

10. Supporting Information S1 and S2 Files have been included

Funding Statement 

The funder Dr. August Wolff GmbH & Co. KG Arzneimittel (Bielefeld, Germany) provided support in the form of salaries for authors [SGdA, LG, MH, and CM] and played an additional role in the study design, decision to publish, and preparation of the manuscript. However, the funder had no role in data collection and analysis. The scientific supervisor of the study [PS] was also sponsored by Dr. August Wolff GmbH & Co. KG Arzneimittel. The specific roles of these authors are articulated in the ‘author contributions’ section. There was no additional external funding received for this study.

Competing Interests Statement

The authors have read the journal’s policy. SGdA, LG, MH, and CM, have the following competing interest: They work for the company Dr. August Wolff GmbH & Co. KG Arzneimittel (Bielefeld, Germany) which is interested in developing products regarding treatment of vulvovaginal atrophy/ vaginal dryness symptoms. The principal investigator of the study [PS] as well as the statistical specialist [TWM] were also sponsored by Dr. August Wolff GmbH & Co. KG Arzneimittel. This does not alter the authors adherence to PLOS ONE policies on sharing data and materials.

Data Availability statement

The minimal data set underlying the results described in this manuscript are fully available. All relevant data are within the paper and its supporting information files.

Supporting information can be found in S1 File, S2 File with the following link https://www.clinicaltrialsregister.eu/ctr-search/trial/2016-002199-28/DE

---

## [Decision Letter · Decision Letter 1]

24 Mar 2022

Vaginal hormone-free moisturising cream is not inferior to an estriol cream for treating symptoms of vulvovaginal atrophy: prospective, randomised study

PONE-D-21-39841R1

Dear Dr. Garcia de Arriba,

We’re pleased to inform you that your manuscript has been judged scientifically suitable for publication and will be formally accepted for publication once it meets all outstanding technical requirements.

Kind regards,

Walid Kamal Abdelbasset, Ph.D.

Academic Editor

PLOS ONE

Additional Editor Comments (optional):

Reviewers' comments:

Reviewer's Responses to Questions

**Comments to the Author**

1. If the authors have adequately addressed your comments raised in a previous round of review and you feel that this manuscript is now acceptable for publication, you may indicate that here to bypass the “Comments to the Author” section, enter your conflict of interest statement in the “Confidential to Editor” section, and submit your "Accept" recommendation.

Reviewer #1: (No Response)

Reviewer #2: All comments have been addressed

2. Is the manuscript technically sound, and do the data support the conclusions?

Reviewer #1: (No Response)

Reviewer #2: Yes

3. Has the statistical analysis been performed appropriately and rigorously? 

Reviewer #1: (No Response)

Reviewer #2: I Don't Know

4. Have the authors made all data underlying the findings in their manuscript fully available?

Reviewer #1: (No Response)

Reviewer #2: Yes

5. Is the manuscript presented in an intelligible fashion and written in standard English?

Reviewer #1: (No Response)

Reviewer #2: Yes

6. Review Comments to the Author

Reviewer #1: (No Response)

Reviewer #2: Dear authors

You answered all the points that I raised and I think that there are no further requirements on my side.

7. PLOS authors have the option to publish the peer review history of their article (what does this mean?). If published, this will include your full peer review and any attached files.

Reviewer #1: No

Reviewer #2: **Yes: **Naser Al-Husban

---

## [Editor Report · Acceptance letter]

4 May 2022

PONE-D-21-39841R1 

Vaginal hormone-free moisturising cream is not inferior to an estriol cream for treating symptoms of vulvovaginal atrophy: prospective, randomised study 

Dear Dr. Garcia de Arriba:

I'm pleased to inform you that your manuscript has been deemed suitable for publication in PLOS ONE. Congratulations! Your manuscript is now with our production department. 

Kind regards, 

on behalf of

Dr. Walid Kamal Abdelbasset 

Academic Editor

PLOS ONE